# Pericarditis in a Child with COVID-19 Complicated by *Streptococcus pneumoniae* Sepsis: A Case Report

**DOI:** 10.3390/v17121567

**Published:** 2025-11-30

**Authors:** Mădălina Maria Merișescu, Mihaela Oroș, Gheorghiță Jugulete, Bianca Borcoș, Larisa Mirela Răduț, Alexandra Totoianu, Anca Oana Dragomirescu

**Affiliations:** 1Faculty of Dentistry, Department of Infectious Diseases, “Carol Davila” University of Medicine and Pharmacy, 050474 Bucharest, Romania; madalina.merisescu@umfcd.ro (M.M.M.); bianca-rybana.bizera@drd.umfcd.ro (B.B.); 2National Institute for Infectious Diseases “Prof. Dr. Matei Bals”, European HIV/AIDS and Infectious Diseases Academy, No. 1 Dr. Calistrat Grozovici Street, 021105 Bucharest, Romania; radutlarisa@gmail.com (L.M.R.); alexandra.totoianu@gmail.com (A.T.); 3Physiology, Department of Preclinical Sciences, Faculty of Medicine, Titu Maiorescu University, No. 67A, Gheorghe Petrașcu Street, 3rd District, 031593 Bucharest, Romania; mihaela.oros@prof.utm.ro; 4Ponderas Academic Hospital, No. 85A, Nicolae G. Caramfil Street, 014142 Bucharest, Romania; 5Faculty of Dentistry, Department of Orthodontics and Dentofacial Orthopaedics, “Carol Davila” University of Medicine and Pharmacy, 050474 Bucharest, Romania; anca.dragomirescu@umfcd.ro

**Keywords:** COVID-19, pediatric sepsis, Streptococcus pneumoniae, pericarditis, otitis media, intravenous immunoglobulin

## Abstract

Background: Pediatric SARS-CoV-2 infection is usually mild, but in rare cases may lead to severe complications. Early recognition and comprehensive management are critical for favorable outcomes. Case Presentation: We present the case of a 2-year-old girl, previously healthy and unvaccinated against *Streptococcus Pneumoniae* (*S. pneumoniae*), who developed SARS-CoV-2 infection and acute otitis media. Initial laboratory evaluation revealed leukocytosis with neutrophilia and increased inflammatory markers. Antiviral and antibiotic treatment was initiated, but she remained febrile, polypneic, and tachycardic. The diagnosis of MIS-C was excluded; there was no involvement of two organs, and infection with *S. pneumoniae* serotype 19 F was identified. Given the unfavorable evolution, corticosteroid therapy and immunoglobulin were instituted, and subsequently, following the antibiogram result, antibiotic therapy was escalated to Meropenem and Linezolid. Clinical and laboratory parameters improved, but pericarditis with a small fluid slide and ECG changes were associated. The evolution was favorable with complete cardiac recovery at 30 days. Conclusion: This case highlights the importance of vigilant assessment for secondary bacterial infections and cardiac complications in pediatric COVID-19. Prompt recognition and targeted treatment are essential, and pneumococcal vaccination remains a fundamental preventive measure. Moreover, the scarcity of literature documenting SARS-CoV-2 infections complicated by pericarditis further underscores the uniqueness of this case and its relevance for specialists in the field.

## 1. Introduction

Since December 2019, the global community has been confronted with an unprecedented pandemic caused by the novel coronavirus SARS-CoV-2, the etiologic agent of coronavirus disease 2019 (COVID-19). Children account for approximately 1–5% of all reported cases, with the proportion requiring hospitalization remaining low. In most pediatric patients, COVID-19 presents with mild or asymptomatic disease [1,2]. SARS-CoV-2 infection may induce immunosuppression in certain individuals, increasing the risk of severe disease and secondary complications. Children with underlying chronic conditions are at higher risk for severe outcomes, which can include acute respiratory distress syndrome, neurological dysfunction, cardiac involvement, or multisystem inflammatory syndrome in children (MIS-C) [3].

Bacterial superinfections represent one of the most significant complications of COVID-19 in both adults and children [4]. In children, there are only a few reports and little information on bacterial coinfection with SARS-CoV-2. In studies conducted in Wuhan, China, only 0.025% of patients were identified with invasive pneumococcal disease in cases of confirmed SARS-CoV-2 infection in adults, and no cases in children. A case of SARS-CoV-2 and *S. pneumoniae* infection complicated by pleurisy and pneumothorax was reported in a child with a favorable outcome [5]. A study from Morocco and Africa found that patients co-infected with SARS-CoV-2 and *Streptococcus pneumoniae* were approximately 2.86 times more likely to experience severe outcomes compared to those infected with *S. pneumoniae* alone [6].

Nasopharyngeal colonization with *Streptococcus pneumoniae* is common in children aged 1–4 years and may serve as a source of invasive infection under conditions of impaired host immunity [7,8]. Pneumococcal conjugate vaccine (PCV) in children has reduced the incidence of invasive lung disease caused by vaccine serotypes. According to a 2025 meta-analysis that included 16 studies, patients with invasive lung disease with serotypes 3, 6A, 11A, 15A, 19F, and 31 were more likely to die [9]. Serotypes 1 and 3 are more often isolated from patients with pneumonia, and serotype 19 F has been associated with the presence of acute otitis media [10]. In a retrospective analysis of invasive pneumococcal disease conducted at the National Institute for Infectious Diseases «Prof. Dr. Matei Balș» between 2011 and 2017, serotype 19F emerged as one of the most prevalent serotypes among the serotyped *Streptococcus pneumoniae* isolates, accounting for approximately 24.2 % of cases [11]. In another study conducted in Romania 2 years before the introduction of vaccination into the national scheme, serotype 19 F was the most frequently found in the nasal discharge of children and was more frequently associated with acute otitis media [12]. Although the frequent presence of serotype 19 F in the pediatric population is noted, the association with pericarditis appears documented in a few international reports and in no report from Romania.

The purpose of this presentation was to illustrate the complexity of a case of pediatric sepsis in a child with SARS-CoV-2 and *S. pneumoniae* serotype 19 F coinfection, subsequently complicated by pericarditis. It represents the first case with this coinfection published in Romania. It is uncertain whether the pericarditis was caused by SARS-CoV-2 or *S. pneumoniae*. Future studies are needed to highlight the serotyping of *S. pneumoniae* in the event of pericarditis as a complication

## 2. Case Presentation

A 2-year-old female child, weighing 20 kg, was transferred in December 2024 from a pediatric hospital to the Clinical Pediatric Infectious Diseases Department IX of the National Institute for Infectious Diseases “Prof. Dr. Matei Balș,” Bucharest, with SARS-CoV-2 infection and sepsis secondary to acute otitis media. She presented with a five-day history of persistent fever, poorly responsive to antipyretics, accompanied by progressive deterioration of her general condition.

The patient was born at term, had normal psychomotor development, and was incompletely vaccinated, notably lacking pneumococcal immunization. She had no previous history of otitis media or hospitalizations, but a month before hospitalization, she required the antibiotic amoxicillin/clavulanic acid for 3 days. This was the only time she took antibiotics. Family history revealed maternal SARS-CoV-2 infection, confirmed by positive antigen testing.

In this case, the diagnosis of SARS-CoV-2 infection was initially established using a rapid antigen detection test and subsequently confirmed by real-time reverse transcription polymerase chain reaction (RT-PCR), in accordance with WHO guidelines [1].

Initial laboratory evaluation revealed leukocytosis with neutrophilia (18,200/mm^3^, 84% neutrophils) and elevated inflammatory markers (CRP 319 mg/L, procalcitonin 2.12 ng/mL). Chest radiography demonstrated accentuation of the peribronchovascular interstitium. Otolaryngologic evaluation confirmed acute bilateral otitis media. Treatment with Remdesivir and Ceftriaxone had been initiated in the pediatric clinic where the patient was initially admitted. She received two doses of Ceftriaxone 100 mg/kg/day, administered once daily, but the clinical course remained unfavorable, with persistent fever, cardio-respiratory instability, somnolence, and irritability, prompting transfer to our clinic.

On admission, vital signs were: BP 108/87 mmHg, HR 172 bpm, RR 34/min, temperature 38.6 °C, SpO_2_ 97% on room air. The child was conscious but in a compromised general state, with pallor, lateral cervical micropolyadenopathy, pharyngeal hyperemia, nasal obstruction, rhinorrhea, productive cough, bilateral bronchial rales, rhythmic heart sounds, supple abdomen, palpable liver at the costal margin, non-palpable spleen, preserved diuresis, normal stools, and no meningeal signs. Laboratory investigations confirmed persistent leukocytosis with neutrophilia (15,400/mm^3^, 80%), microcytic hypochromic anemia (Hb 9.9 g/dL), and elevated inflammatory markers (CRP 85 mg/dL, fibrinogen 990 mg/dL, procalcitonin 1.62 ng/mL) See Table 1. Multiplex PCR of the upper respiratory tract was performed, which identified infection with *Streptococcus pneumoniae* serotype 19 F. Upon ENT reevaluation, the diagnosis of right perforated otitis media is established, and ear secretion is collected. Culture from the ear discharge also revealed *S. pneumoniae*, and antibiotic susceptibility testing revealed penicillin resistance, with sensitivity to trimethoprim-sulfamethoxazole, linezolid, fluoroquinolones, and vancomycin. Nasal swab cultures grew *Streptococcus pneumoniae,* and blood cultures were negative. Antibiotic therapy was escalated to Meropenem and Linezolid, and immunoglobulin treatment was introduced for 3 days, and corticosteroid treatment with Dexamethasone for six days. See Table 2.

The evolution was favorable with remission of fever, but on the fifth day of hospitalization, the patient developed precordial pain and irritability; echocardiography revealed a small circumferential pericardial effusion (0.5 cm) with mild mitral regurgitation. Considering the child’s age, the ECG showed negative T waves, but pathologically, tachycardia with ST-segment depression was also evident. See Figure 1. Cardiac markers (troponin, CK, and NT-proBNP) were collected during hospitalization with normal values, excluding the diagnosis of myocarditis. Acute pericarditis was diagnosed according to international guidelines, requiring at least two of the following criteria: typical chest pain, suggestive electrocardiographic changes, pericardial friction rub, or echocardiographic evidence of pericardial effusion [3]. Pericardial effusions were classified echocardiographically as small (<10 mm), moderate (10–20 mm), or large (>20 mm). See Figure 2 [4].

Pediatric sepsis was diagnosed based on the presence of at least two criteria defined by the Pediatric Systemic Inflammatory Response Syndrome (pSIRS) and documentation of infection in the ear culture. The pSIRS criteria include age-adjusted tachycardia or bradycardia, tachypnea, leukocytosis or leukopenia, and hypotension [2,13,14].

Identification of a bacterial agent and fulfillment of the criteria for bacterial sepsis argue against the diagnosis of multisystem inflammatory syndrome (MIS-C). Patients with MIS-C are characterized by the presence of at least 2 affected organs (mucocutaneous involvement, hypotension, myocarditis, pericarditis, evidence of coagulopathy, acute gastrointestinal problems) in the absence of a bacterial etiology and, usually, 4–6 weeks after infection with SARS-CoV-2 [15].

## 3. Treatment and Outcome

The patient received antiviral therapy with Remdesivir for five days, broad-spectrum antibiotics with Meropenem and Linezolid for seven days, corticosteroids administered at an initial dose of 6 mg/day, corresponding to 0.3 mg/kg/day, followed by a progressively tapered regimen. A total of six days of corticosteroid therapy was administered, along with a three-day course of intravenous immunoglobulin (IVIG), both of which were well tolerated without adverse reactions. The arrows in Table 2 indicated the time at which the treatment was initiated and its duration. See Table 2

Clinical evolution was favorable, with resolution of fever. At discharge, ST-segment depression persisted, but tachycardia and chest pain resolved. At the 30-day follow-up, physical examination was unremarkable. Laboratory tests showed only mild hypochromic microcytic anemia. Otorhinolaryngological evaluation was normal, and follow-up echocardiography did not demonstrate pericardial effusion, and the ECG normalized. The patient remains under outpatient surveillance, with periodic evaluations.

## 4. Discussion

Children constitute a relatively small proportion of total COVID-19 cases compared to adults. A study conducted in the United Kingdom demonstrated that less than 1% of pediatric COVID-19 cases result in hospitalization [16]. Despite the generally favorable clinical course, pediatricians remain vigilant regarding the potential development of severe complications, including septic shock or multisystem inflammatory syndrome in children (MIS-C) associated with SARS-CoV-2 infection. In a cohort study conducted in Geneva involving 57 children with confirmed SARS-CoV-2 infection, one case of septic shock was reported, while two additional patients met the criteria for multiple organ dysfunction syndrome (MODS) [16,17].

The present case describes a pediatric patient who developed sepsis secondary to otitis media caused by *Streptococcus pneumoniae* serotype 19 F in the context of SARS-CoV-2–induced immunosuppression. On the tenth day of illness and the fifth day of hospitalization, the child exhibited acute chest pain and a small circumferential pericardial effusion. Literature indicates that SARS-CoV-2 can directly invade pericardial and myocardial tissues, triggering local inflammation and impairing cardiac function. In one study of 530 hospitalized patients, pericardial effusion was observed in 75 cases (14%), whereas only 17 patients (3.2%) fulfilled the criteria for acute pericarditis. Prospective studies in critically ill COVID-19 patients have reported the prevalence of pericardial effusion ranging from 43% to 90% [18]. Small-volume pericardial effusions are frequently encountered, particularly during the post-acute phase of COVID-19 [19,20]. Although most pediatric patients recover fully from viral pericarditis, the literature recommends follow-up with echocardiography and cardiac biomarkers (troponin, NT-proBNP) to detect potential late-onset myocardial involvement, including ventricular dysfunction or arrhythmias [21,22].

Secondary bacterial infections and co-infections represent significant risk factors for adverse outcomes in patients with SARS-CoV-2 infection. Alshaikh et al., in a meta-analysis of 20 studies published between December 2019 and June 2021, reported a bacterial co-infection prevalence of 5.62% among patients with COVID-19. The most frequently identified pathogens included *Streptococcus pneumoniae*, *Staphylococcus aureus*, and *Haemophilus influenzae*. Specifically, *S. pneumoniae* was most commonly isolated from the respiratory tract [23]. In a cohort of 120 hospitalized patients, percentages between 20% and 60% of the frequency of coinfection between *S. pneumoniae* and SARS-CoV-2 were reported [24].

Lack of pneumococcal immunization was a critical risk factor for the development of invasive bacterial infection in this case. Evidence demonstrates that vaccination with PCV13 or PCV15 significantly reduces the incidence of invasive pneumococcal disease, even in the context of viral co-infections [11,21,24]. Lack of pneumococcal immunization was a critical risk factor for the development of invasive bacterial infection in this case [24].

There is limited data regarding the incidence of bacterial sepsis among pediatric patients with SARS-CoV-2 infection. Severe COVID-19 or early-onset MIS-C may be difficult to distinguish from bacterial sepsis, as both conditions can present with multi-organ dysfunction. In this context, organ dysfunction associated with COVID-19 may be considered a pediatric sepsis phenotype or septic shock. Tripathi et al., in 2025, reported for the first time the prevalence of sepsis among pediatric SARS-CoV-2 patients using the Phoenix Sepsis Score, identifying 18.8% of patients meeting sepsis criteria and 9.7% meeting septic shock criteria in a cohort of 1731patients [20,25]. In the present case, the 2005 International Pediatric Sepsis Consensus Conference (IPSCC) criteria were applied, confirming *S. pneumoniae* as the etiologic agent of sepsis, while the clinical features did not fulfill the MIS-C diagnostic criteria. Identification of a bacterial infection excludes the diagnosis of MIS-C [26].

The pathogenetic relationship between SARS-CoV-2 and secondary pneumococcal infection

SARS-CoV-2 can alter both innate and adaptive immunity, increasing susceptibility to secondary bacterial infections. Studies indicate impaired alveolar macrophage function and reduced naive B-cell responses, which may facilitate bacterial colonization in the respiratory tract [22,27]. Early in SARS-CoV-2 infection, the virus primarily infects type 2 pneumocytes, and this condition can increase the risk of secondary bacterial infections due to a weakened lung immune response [28]. Respiratory viral infections, including SARS-CoV-2, increase susceptibility to acute otitis media by promoting inflammation of the nasopharyngeal mucosa, impairing mucociliary function, leading to accumulation of secretions, and reducing the efficiency of local immune defenses [29]. During the SARS-CoV-2 Omicron variant pandemic, the incidence of acute otitis media increased by approximately 15% compared to periods without viral infection. The onset of otitis typically occurred between 7 and 9 days following SARS-CoV-2 infection. Upper respiratory tract PCR analysis also encompassed influenza virus, a pathogen frequently implicated in severe pediatric cases and known to predispose to secondary bacterial infections, particularly those caused by *Streptococcus pneumoniae* [27,30].

Children colonized with bacteria in the context of SARS-CoV-2 infection may manifest bacterial disease. Although bacteria do not directly influence viral replication, they may enhance viral adaptation, increase virion stability, and facilitate cellular infection [31].

Furthermore, it is well documented that co-infections exacerbate the severity of acute SARS-CoV-2 infection by enhancing viral adaptation, may increase virion stability, and facilitate cellular infection. In a study of 120 patients, those co-infected with SARS-CoV-2 had higher rates of ICU admission and mortality compared to those not co-infected [32]. Conversely, viral infections not only create favorable conditions for bacterial colonization but can also enhance bacterial virulence, potentially leading to more severe disease outcomes [6,26].

Integrated Management of Co-infections

Combination therapy—including antivirals, corticosteroids, IVIG, and appropriately targeted antibiotics—is appropriate in severe cases. De-escalation strategies guided by culture and antibiogram data are associated with reduced antibiotic resistance and adverse events. The use of corticosteroids is indicated in children with SARS-CoV-2 and MIS-C infection, but its use in children with sepsis is controversial. In our case, the use of corticosteroid therapy in a child with sepsis was successful.

Early administration of intravenous immunoglobulin (IVIG) within the first 48 h in children with severe or critical COVID-19 has been shown to increase survival rates and reduce the length of hospital stay. Additionally, high-dose IVIG therapy (2 g/kg within the first 14 days of illness) has demonstrated improved clinical outcomes in this patient population [21,22]. The immunomodulatory effect in patients with elevated inflammatory markers, IL-6, CRP, has been proven in many previous studies. Immunoglobulins may also have effects on neutralizing pathogens, activating phagocytosis, activating and modulating complement, or restoring immunoglobulin deficiency in patients with hypogammaglobulinemia and sepsis. There are limited studies or case reports that recommend the use of immunoglobulins in children with pediatric sepsis. Our case supports the beneficial effect of the use of immunoglobulins in a child with pneumococcal sepsis [33].

For future research directions, prospective multicenter studies are needed to evaluate the prevalence of bacterial sepsis and cardiovascular complications in pediatric COVID-19 and to assess the impact of pneumococcal vaccination on disease severity.

## 5. Conclusions

This case underscores the complexity of pediatric SARS-CoV-2 infection and demonstrates that even previously healthy, immunocompetent children may develop severe complications, particularly when bacterial co-infections are present. The absence of pneumococcal immunization in this patient highlights the critical role of vaccination in preventing invasive bacterial disease, especially in the context of viral infections such as COVID-19.

The integration of cardiological evaluation with bacteriological investigations facilitated the differentiation of sepsis from MIS-C, enabling the correct orientation of therapeutic management.

Cardiovascular complications, including pericardial effusion, warrant careful monitoring through echocardiography and biomarker assessment to ensure complete recovery and detect potential late-onset sequelae

This case also highlights the broader implications for public health and research, indicating the need for enhanced surveillance of bacterial co-infections in pediatric COVID-19 and further studies to identify risk factors, optimize management strategies, and clarify the protective role of vaccination.

In conclusion, while SARS-CoV-2 infection in children is generally mild, it can rarely result in complications, and preventive strategies, early detection, and comprehensive treatment are crucial to minimize morbidity and mortality.

## Figures and Tables

**Figure 1 viruses-17-01567-f001:**
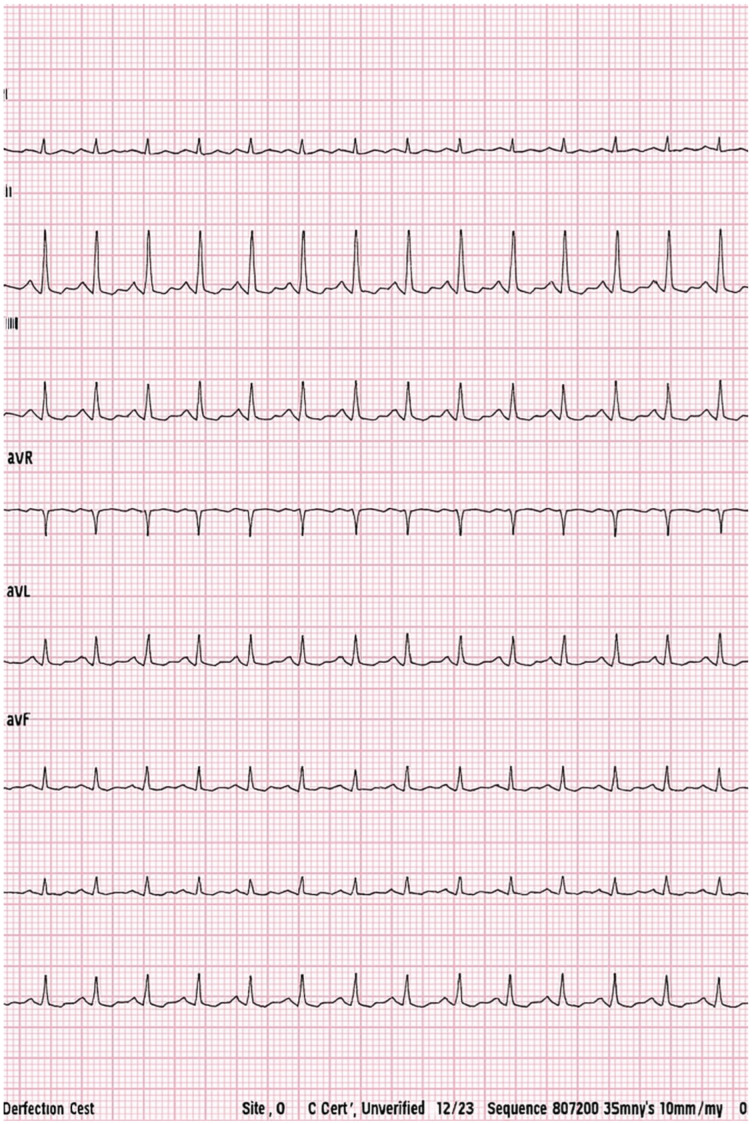
Tachycardia and ST-segment depression on ECG. Considering the child’s age, the ECG showed negative T waves, but pathologically, tachycardia with ST-segment depression was also evident.

**Figure 2 viruses-17-01567-f002:**
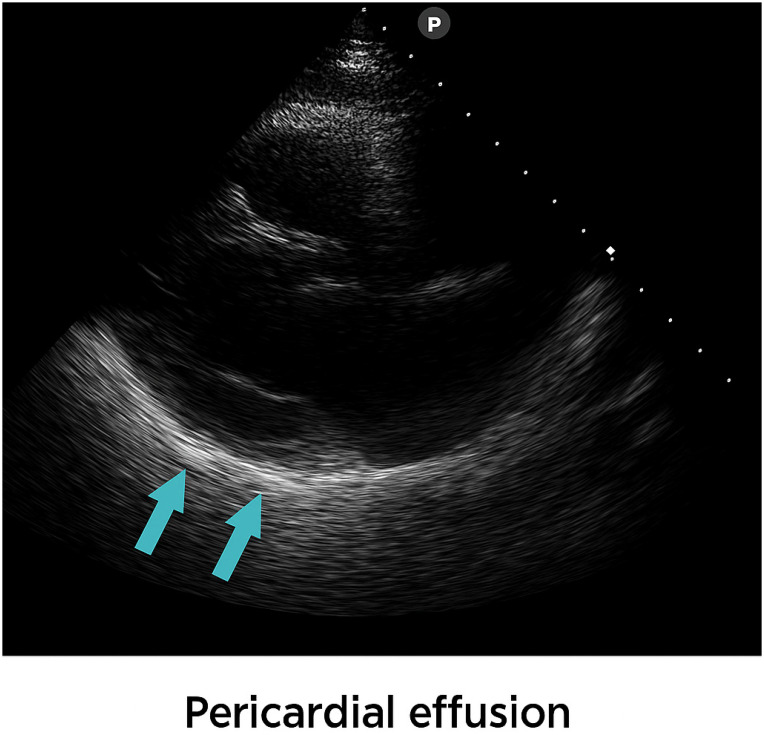
Picture 1 with pericardial effusion. Picture 2 Echocardiographic image demonstrating a small circumferential pericardial effusion (0.5 cm) with mild mitral regurgitation. The arrows in the images highlight the pericardial fluid.

**Table 1 viruses-17-01567-t001:** Evolution of Laboratory Parameters.

Hospital Day	1	4	7	10	Reference Range
WBC (/mm^3^)	7500	7700	10,430	17,900	6000–17,000
Neutrophils (/mm^3^)	2100	2150	3160	4200	1800–8000
Lymphocytes (/mm^3^)	4000	4300	6290	12,100	3000–9500
Hemoglobin (g/dL)	9.7	9.9	10.5	10.7	12–15
CRP (mg/L)	319	85	23	2	0–5
Fibrinogen (mg/dL)	890	630	366	254	200–400
PCT (ng/mL)	2.12	1.62	0.11	0.06	0–0.5
Troponin (ng/mL)	–	0.03	0.03	–	0–0.04

**Table 2 viruses-17-01567-t002:** Clinical, paraclinical, and treatment chronologically. The arrows in the text highlight the duration of the treatment and the point at which it was initiated.

Hospital Day	1	2	3	4	5	6	7	8	9	10
Fever (>38 °C)	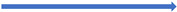							
Chest pain					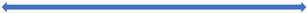		
PCR Biofire			*S.pneumoniae*Serotype 19 F							
Otic secretion			Sampling			*S.pneumoniae*				
Pericardial fluid					0.5 cm		0.3 cm			
Remdesivir	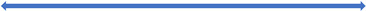					
Antibiotics	Ceftriaxone	Meropenem + Linezolid	
CorticosteroidDexamethasone	-	-	6mg/zi	6 mg/zi	4 mg/zi	4mg/zi	2 mg/zi	2 mg/zi	-	-
IVIG			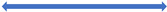					

## Data Availability

The original contributions presented in this study are included in the article. Further inquiries can be directed to the corresponding author(s).

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
