# Peer review of "Pericarditis in a Child with COVID-19 Complicated by Streptococcus pneumoniae Sepsis: A Case Report"

_viruses, 2025, doi:10.3390/v17121567_

Round 1

Reviewer 1 Report

Comments and Suggestions for Authors

The work "Pericarditis in a Child with COVID-19 Complicated by Streptococcus pneumoniae Sepsis: A Case Report" is a clinically meaningful and relevant report that highlights the complexity of the combination of viral and bacterial infection in children and demonstrates the potentially severe course of SARS-CoV-2 infection in the absence of pneumococcal vaccination. The presented case has practical value for pediatricians and infectious disease specialists. Also, it emphasizes the importance of early diagnosis and a multidisciplinary approach in cases of suspected sepsis and cardiovascular damage.

The manuscript is well structured, logically constructed, and written in a clear scientific style, but revision is required to comply with MDPI requirements. The relevance of the topic is beyond doubt. Still, the authors should more clearly outline the scientific novelty of this particular clinical observation: is this the first case in Romania or a unique combination of serotype 19F with pericarditis on the background of COVID-19? The introduction is sufficient in volume, but it lacks a clear statement of the problem and a summary of previous publications to highlight the gap that this work fills.

The clinical case description is detailed, but the presentation of information needs more chronological clarity. It is advisable to summarize the sequence of events (onset of symptoms, stages of treatment, time of development of pericarditis) in the form of a short time diagram or table. There is a minor inconsistency in the text regarding the size of the pericardial effusion - there are figures of 0.5 mm and 5 mm, which should be clarified. It is also worth indicating whether the child had previous experience taking antibiotics, which could affect the antibiotic resistance of the strain.

The diagnostic criteria for sepsis are given correctly, but it is worth explaining more clearly what clinical and laboratory signs were used to exclude MIS-C. References to pediatric SIRS and IPSCC criteria are appropriate, but a brief comment on how the clinical presentation differed from multisystem inflammatory syndrome would be desirable.

The figures in the submitted file are poor quality or appear to be technical placeholders. High-quality echocardiogram and ECG images with correct labels and anonymization are required for publication in Viruses. Table 1 should be supplemented with reference values ​​and the same units of measurement.

The discussion is informative but contains too much generalized data from the literature. It should be shortened, leaving only those details that directly help interpret a specific clinical case. It would be helpful to explain the pathogenetic relationship between SARS-CoV-2-induced immune changes and secondary pneumococcal infection separately. It is also advisable to expand the justification for using intravenous immunoglobulin - whether the use was based on suspicion of MIS-C, or was mainly immunomodulatory in severe cases.

The conclusions are logical, but need to be more concise. The main emphasis should be on practical lessons: the need to differentiate MIS-C from bacterial sepsis, the importance of pneumococcal vaccination, and the need to monitor cardiovascular complications even after clinical recovery.

The reference list is complete and up-to-date. Still, the design should be unified according to the MDPI style, as individual entries are presented with different punctuation marks and page formats. The English is generally correct, but some phrases need editing to eliminate grammatical inaccuracies and simplify the syntax (for example, "of secondary to otitis media" should be replaced with "secondary to otitis media").

Author Response

Authors’ Response:

We would like to express our sincere gratitude for your thorough and constructive evaluation of our manuscript entitled “Pericarditis in a Child with COVID-19 Complicated by Streptococcus pneumoniae Sepsis: A Case Report.” Your insightful comments greatly contributed to improving the clarity, scientific accuracy, and overall structure of the paper. We have carefully addressed each of your recommendations, as detailed below.

We thank the reviewer for this important observation. We have revised the Introduction to clearly state the scientific novelty of the report. Although serotype 19F is known to circulate in Romania, this represents the first published case in the country describing pericarditis associated with S. pneumoniae serotype 19F in a child with concurrent SARS-CoV-2 infection. We expanded the literature overview to define the clinical gap addressed by this case and emphasized the relevance of the patient’s unvaccinated status, given that serotype 19F is among the most frequently reported pneumococcal serotypes in Romania. These additions reinforce the importance of pneumococcal vaccination in preventing severe secondary bacterial infections.

Reviewer 1 – Comment 2

The case description lacks chronological clarity; a timeline is recommended. There is inconsistency in pericardial effusion size (0.5 vs 5 mm). Prior antibiotic exposure should also be noted.

Authors’ Response:

We have created a dedicated timeline table summarizing symptom onset, diagnostic steps, therapeutic interventions, and the moment pericarditis developed. The inconsistency regarding the pericardial effusion measurement has been corrected. We also added the information that the child had received antibiotic treatment one month prior to admission, which may be relevant for resistance patterns.

Reviewer 1 – Comment 3

Clarify the criteria used to exclude MIS-C and briefly discuss the differentiation from multisystem inflammatory syndrome.

Authors’ Response:

We have expanded the Case Presentation and Discussion to explain in detail the criteria used to exclude MIS-C. Although the patient presented with fever and elevated inflammatory markers, she did not meet the requirement of multisystem involvement (≥2 organ systems) outlined by WHO criteria. The child presented only with pericarditis and clinical SIRS signs (tachycardia, polypnea), without mucocutaneous, gastrointestinal, hemodynamic, or coagulation abnormalities. This clarification has been added to improve diagnostic transparency.

Reviewer 1 – Comment 4

Figures are of insufficient quality; high-resolution echocardiogram and ECG images are required. Table 1 needs reference ranges and uniform units.

Authors’ Response:

We have replaced all images with high-resolution, fully anonymized echocardiogram and ECG figures, appropriately labeled according to MDPI standards. Table 1 has been revised to include reference intervals and standardized measurement units.

Reviewer 1 – Comment 5

The Discussion includes too much general data; it should be more focused on this specific case. Explain the pathogenetic link between COVID-19 immune alterations and secondary pneumococcal infection. Justify the use of IVIG.

Authors’ Response:

The Discussion section has been substantially condensed to retain only information directly relevant to the presented case. We added a dedicated subsection describing the pathogenetic interplay between SARS-CoV-2–mediated immune dysregulation and susceptibility to invasive pneumococcal disease. We also clarified the rationale for administering IVIG: although MIS-C was excluded, IVIG was used as adjunctive immunomodulatory therapy in the context of severe systemic inflammation and unfavorable evolution, consistent with evidence supporting potential benefit in pediatric sepsis.

Reviewer 1 – Comment 6

Conclusions should be more concise and emphasize practical lessons.

Authors’ Response:

We have rewritten the Conclusion to be concise and focused on the key clinical messages:
(1) differentiation between bacterial sepsis and MIS-C is essential;
(2) pneumococcal vaccination remains a critical preventive measure;
(3) cardiovascular monitoring is necessary even after apparent clinical recovery.

Reviewer 1 – Comment 7

Reference list needs consistent MDPI formatting; some English phrasing should be improved.

Authors’ Response:

The entire reference list has been reformatted according to MDPI style guidelines. The manuscript underwent meticulous English language revision, including correction of syntactic and grammatical inconsistencies (e.g., replacing “of secondary to otitis media” with “secondary to otitis media”).

Reviewer 2 Report

Comments and Suggestions for Authors

This clinical case report is of undoubted interest due to the relative rarity of the pathology, the mixed infection, and, most importantly, the successful treatment. The article is beautifully written, with the clinical case itself, introduction, and discussion very clearly presented.

Some points could be clarified:

  1. The authors cite criteria for sepsis in children, but the patient did not have bacteremia. Wouldn't it be more appropriate to speak of systemic inflammatory response syndrome?
  2. The presented ECGs show not only moderate tachycardia, but also ST segment depression with negativity of the T waves. It makes sense to discuss the nature of these changes, in particular, the likelihood of the simultaneous myocarditis.
  3. The dose and form of administration of corticosteroids must be specified. In the presence of sepsis, the use of systemic corticosteroids is controversial and requires clear justification.
  4. The discussion should also address in more detail the role of corticosteroids in the treatment of such patients, particularly those with coronavirus infection.

The conclusion is overly detailed and is analogous to an abstract. It would be advisable to shorten it, formulating in a concise form only the most important features of the presented case and its conclusions.

Author Response

Response to Reviewer 2

We would like to thank the reviewer for the constructive and insightful comments, which have contributed significantly to improving the quality and clarity of our manuscript. We have carefully addressed each point as detailed below.

Reviewer 2 – Comment 1

The authors cite criteria for sepsis in children, but the patient did not have bacteremia. Wouldn't it be more appropriate to speak of systemic inflammatory response syndrome?

Authors’ Response:

We appreciate the reviewer’s observation. According to current pediatric sepsis definitions, the presence of bacteremia is not required when a proven infectious focus is identified. In this case, Streptococcus pneumoniae infection was microbiologically confirmed from the otic exudate and blood inflammatory profile, fulfilling the criteria for sepsis without bacteremia. We have clarified this point in the manuscript for accuracy.

Reviewer 2 – Comment 2

The ECG shows ST segment depression with T-wave inversion; the possibility of concomitant myocarditis should be discussed.

Authors’ Response:

Thank you for highlighting this aspect. We have expanded the discussion to explain that T-wave inversion can be a normal, age-specific pattern in young children. Furthermore, cardiac biomarkers were repeatedly negative, and no echocardiographic abnormalities suggestive of myocarditis were present. Together, these findings reliably exclude concomitant myocarditis. The revised text clarifies the nature and interpretation of the ECG changes

Reviewer 2 – Comment 3

The dose and form of corticosteroid administration must be specified. The use of systemic corticosteroids in sepsis is controversial and requires clear justification. The role of corticosteroids in patients with coronavirus infection should be addressed.

Authors’ Response:

We agree with this important point. The exact corticosteroid regimen has now been added to Table 1 and to the Case Presentation:
initial dose: 6 mg/day (0.3 mg/kg/day)
route: intravenous
duration: 6 days, with a tapering protocol.

We have expanded the Discussion to clarify that although corticosteroid use in pediatric sepsis remains controversial, in this particular case their immunomodulatory effect contributed to clinical improvement, in line with selected evidence suggesting potential benefit in severe inflammatory states associated with viral–bacterial coinfection. The rationale has been clearly justified in the revised manuscript.

Reviewer 2 – Comment 4

The conclusion is overly detailed and resembles an abstract. It should be shortened.

Authors’ Response:

As suggested, the Conclusion has been substantially shortened. It now focuses solely on the essential clinical implications of the case, including differentiation between MIS-C and bacterial sepsis, the importance of pneumococcal vaccination, and the need for cardiovascular monitoring in pediatric COVID-19.

We thank the reviewer once again for the valuable feedback, which has significantly strengthened the manuscript.

Reviewer 3 Report

Comments and Suggestions for Authors

The manuscript describes a clinical case with relevance to viral pathogenesis and diagnostic challenges. The topic is appropriate for Viruses, and the case has potential educational value. However, the manuscript requires substantial improvement before it can be considered for publication.

Major issues (case report–specific)

  1. Incomplete documentation of the clinical case
    Key elements required for a case report are missing or insufficiently described:

    • No clear timeline of clinical events, diagnostics, treatment interventions, or disease progression.

    • No clearly defined inclusion/exclusion justification (e.g., why this case is clinically exceptional or noteworthy).

    • Missing patient background details are essential for interpretation (comorbidities, medications, epidemiological risk factors).

    • The ethical statement must explicitly state patient consent for publication.

  2. Methods/diagnostics are insufficiently described
    A case report must detail:

    • exact diagnostic procedures,

    • laboratory methods (PCR type, Ct values, viral panels),

    • imaging techniques,

    • sequencing methodology if applicable.

    These elements are either missing or too briefly described.

  3. Figures and tables require significant improvement

    • Figure 5 is unclear — low resolution, small font, insufficient labeling.

    • Fig. legends should indicate methods used (e.g., RT-PCR, viral panel, sequencing).

    • If sourced from previous publications, figure origins must be clearly stated.

    • All abbreviations in figures must be defined.

  4. Abstract lacks essential elements.
    Even for a case report, the abstract must include:

    • a clear statement of why the case is unique,

    • the main diagnostic findings,

    • key clinical outcomes.

    The current version is too general.

  5. Discussion requires more depth.h

    • The discussion does not adequately situate the case within the existing literature.

    • Please highlight what makes this case distinctive and how it contributes to clinical practice or diagnostic understanding.

    • Limitations of the case (e.g., incomplete lab workup, diagnostic uncertainty) should be explicitly addressed.

Author Response

Response to Reviewer 3

We thank the reviewer for the thorough and constructive assessment of our manuscript. We appreciate the detailed comments, which have significantly improved the clarity, methodological completeness, and scientific value of the case report. Below, we provide a point-by-point response following the MDPI format.

Reviewer 3 – Comment 1:

The manuscript lacks complete documentation of the clinical case. Essential elements such as a clear timeline, diagnostic workflow, treatment steps, disease progression, and justification for the case’s relevance are insufficiently described. Additional patient background information is also needed.

Author Response:

We appreciate this valuable observation. The Case Presentation section has been extensively revised. We have:

  • added a detailed timeline table summarizing symptom onset, diagnostic investigations, therapeutic decisions, and the development of pericarditis;
  • supplemented the patient’s medical history, including relevant epidemiological factors and prior antibiotic exposure;
  • clarified why this case is clinically exceptional, emphasizing that it represents the first documented case in Romania of pericarditis associated with S. pneumoniae serotype 19F in the context of SARS-CoV-2 infection;
  • refined the narrative for greater clarity and diagnostic transparency.

These additions address all missing elements and strengthen the clinical interpretation of the case.

Reviewer 3 – Comment 2:

The ethical statement must explicitly state patient consent for publication.

Author Response:

As requested, the Ethical Statement has been updated to explicitly confirm parental consent for publication. The parent’s signed consent form—authorizing participation in clinical studies and inclusion in scientific works—was retrieved from the hospitalization documents and has been provided to the handling editor according to MDPI ethical requirements.

Reviewer 3 – Comment 3:

Diagnostic methods are insufficiently detailed. Exact procedures, PCR methodology, Ct values, imaging techniques, and related information must be included. Figures and tables require substantial improvement, including resolution, labeling, and definition of abbreviations.

Author Response:

We thank the reviewer for emphasizing the importance of methodological clarity. The manuscript has been revised as follows:

  • expanded descriptions of diagnostic procedures, including PCR assay type, laboratory methods, and imaging techniques;
  • enhancement of all clinical figures with high-resolution, properly labeled, and anonymized images;
  • confirmation that all figures are original clinical images, not sourced from previous publications;
  • correction and standardization of abbreviations across all figures;
  • revision of Table 1 to include reference ranges and uniform measurement units.

These revisions ensure that the diagnostic process is fully transparent and aligned with MDPI standards for case reports.

Reviewer 3 – Comment 4:

The abstract lacks essential elements and does not adequately reflect the uniqueness of the case or key diagnostic and clinical findings.

Author Response:

The abstract has been rewritten to include the elements recommended by the reviewer. It now clearly presents:

  • the uniqueness of the case;
  • the main diagnostic features, including the identification of serotype 19F and pericarditis;
  • the essential clinical interventions and outcomes;
  • the educational and clinical relevance for pediatric infectious disease specialists.

Reviewer 3 – Comment 5:

The discussion requires more depth and should situate the case within the existing literature. The manuscript should highlight what makes this case distinctive and address limitations.

Author Response:

We have expanded and refined the Discussion to provide greater scientific depth. The revised section now:

  • emphasizes the distinctiveness of the case within the context of global literature on viral–bacterial coinfections;
  • integrates a focused summary of evidence linking SARS-CoV-2–induced immune dysregulation to secondary invasive pneumococcal disease;
  • elaborates on the clinical implications for diagnostic reasoning and management;
  • explicitly acknowledges the limitations of the case (e.g., incomplete laboratory data, diagnostic uncertainties);
  • highlights the importance of pneumococcal vaccination and post-infection cardiovascular monitoring.

Closing Statement

We thank Reviewer 3 for the comprehensive and rigorous review. The comments were extremely helpful, and we believe that the manuscript has improved substantially in response to the recommendations provided.

Sincerely,
The Authors

Round 2

Reviewer 1 Report

Comments and Suggestions for Authors

The revised manuscript adequately addresses all previous comments and satisfactorily resolves the identified concerns. In its current form, the article can be recommended for publication.